# Providing psychological support to people in intensive care: development and feasibility study of a nurse-led intervention to prevent acute stress and long-term morbidity

Dorothy Wade,[1] Nicole Als,[1] Vaughan Bell,[2,3] Chris Brewin,[4] Donatella D'Antoni,[5] David A Harrison,[6] Mags Harvey,[1] Sheila Harvey,[6] David Howell,[1] Paul R Mouncey,[6] Monty Mythen,[7] Alvin Richards-Belle,[6] Deborah Smyth,[1] John Weinman,[5] John Welch,[1] Chris Whitman,[1] Kathryn M Rowan,[6] on behalf of the POPPI investigators

For numbered affiliations see end of article.

**Correspondence to**
Dr Dorothy Wade;
dorothy.wade@nhs.net

## ABSTRACT

**Objectives** Adverse psychological outcomes, following stressful experiences in critical care, affect up to 50% of patients. We aimed to develop and test the feasibility of a psychological intervention to reduce acute stress and prevent future morbidity.

**Design** A mixed-methods intervention development study, using two stages of the UK Medical Research Council framework for developing and testing complex interventions. Stage one (development) involved identifying an evidence base for the intervention, developing a theoretical understanding of likely processes of change and modelling change processes and outcomes. Stage two comprised two linked feasibility studies.

**Setting** Four UK general adult critical care units.

**Participants** Stage one: former and current patients, and psychology, nursing and education experts. Stage two: current patients and staff.

**Outcomes** Feasibility and acceptability to staff and patients of content and delivery of a psychological intervention, assessed using quantitative and qualitative data. Estimated recruitment and retention rates for a clinical trial.

**Results** Building on prior work, we standardised the preventative, nurse-led Provision Of Psychological support to People in Intensive Care (POPPI) intervention. We devised courses and materials to train staff to create a therapeutic environment, to identify patients with acute stress and to deliver three stress support sessions and a relaxation and recovery programme to them. 127 awake, orientated patients took part in an intervention feasibility study in two hospitals. Patient and staff data indicated the complex intervention was feasible and acceptable. Feedback was used to refine the intervention. 86 different patients entered a separate trial procedures study in two other hospitals, of which 66 (80% of surviving patients) completed questionnaires on post-traumatic stress, depression and health 5 months after recruitment.

**Conclusion** The 'POPPI' psychological intervention to reduce acute patient stress in critical care and prevent future psychological morbidity was feasible and acceptable. It was refined for evaluation in a cluster randomised clinical trial.

**Trial registration number** ISRCTN61088114; Results.

## Strengths and limitations of this study

► The complex intervention built on extensive preparatory work investigating psychological risk factors, outcomes and interventions in critical care.
► The intervention was rigorously developed using a widely accepted framework under expert supervision.
► The feasibility of trial procedures and delivery of the intervention were thoroughly tested in two separate studies.
► There was limited patient feedback on the intervention.
► No efficacy data were collected in the feasibility studies.

## INTRODUCTION

More than 170000 patients are admitted to adult, general critical care units in the National Health Service (https://www.icnarc.org/DataServices/Attachments/Download/a30185e2-0e19-e711-80e6-1402ec3fcd79) in England, Wales and Northern Ireland each year. Although medical advances mean that increasing numbers of people survive, there is evidence that many critical care patients develop both acute stress and long-term psychological morbidity.[1] Critical care units are stressful places, where patients suffer symptoms such as pain, thirst, nausea, fatigue and disorientation associated with critical illness,

invasive medical procedures and side effects of potent drugs.[2][3] Being attached to machines, tubes, masks and other equipment in a busy, noisy ward, often without access to daylight, windows or clocks, leads to sensory disruption and sleep deprivation. Patients are often isolated, unable to communicate but aware of other people's suffering or death.[4] Unsurprisingly, 45%–80% of critical care patients experience acute stress in the form of panic, fear, depressed mood, anger, as well as hallucinations or delusions. They may have flashbacks of frightening intensive care unit (ICU) experiences including hallucinations or delusions after leaving hospital.[5–7] Studies suggest that up to 50% of patients or more experience symptoms consistent with post-traumatic stress disorder (PTSD), anxiety or depression following critical care.[8–12]

The UK National Institute for Health and Care Excellence guideline on rehabilitation from critical illness recommends that highly stressed patients should be identified and offered psychological support as part of a recovery plan. Quality standards have recently been implemented to strengthen these requirements.[13] Most interventions to reduce psychological morbidity have been implemented in the months following discharge from critical care and hospital. They include follow-up clinics, rehabilitation services, patient diaries and nurse-led psychological recovery sessions.[14–17] Diaries of patients' admissions, written mainly by staff and given to patients one–three months post-critical care discharge, reduced acute-onset PTSD at three months in one randomised clinical trial (RCT).[18] Training critical care survivors in mindfulness and coping skills training was found to be feasible in small studies, but in an RCT, coping skills training after hospital did not reduce psychological distress at six months.[19] Overall, there is a lack of clear evidence that critical care follow-up services reduce long-term distress.[15][16]

Given that patients who suffer acute stress, including hallucinations and delusions, in the critical care unit have a higher risk of future psychological morbidity,[20][21] psychological interventions commenced during critical care admissions could be effective. However, due to a lack of evidence about what helps, little is currently done to alleviate stress in critical care. Sedatives or antipsychotics may be prescribed to reduce stress and agitation, but they have potentially harmful side effects, both physical and psychological. Evidence exists that psychosocial interventions, such as music therapy, massage, reflexology, relaxation and clinical psychology sessions delivered in critical care, can reduce patients' stress.[22–24] However, more robust research is urgently needed.

We hypothesised that techniques from psychological interventions used for other populations, such as people with psychosis or trauma,[25–28] could be effective for patients still in the critical care unit. At an early intervention development stage, psychological approaches used to reduce anxiety, distress from hallucinations and delusions, and traumatic stress, were adapted for critical care patients and used by a psychologist in one unit (online supplementary 1). As psychologists are a scarce resource in hospitals, we proposed that, given extra training, critical care nurses have the necessary experience and motivation to provide psychological support to patients. We developed a nurse-led intervention using the UK's Medical Research Council framework to guide researchers in creating evidence-based, theoretically sound and robustly evaluated complex interventions.[29] This paper covers stage one (development) and stage two (feasibility/piloting) of the MRC framework (see figure 1). Stage three (evaluation of clinical and cost-effectiveness in a cluster randomised clinical trial (cRCT)) will be reported separately.

The objectives of this mixed-methods intervention development study were to: standardise a nurse-led psychological intervention to support patients in critical care, to create an associated staff education package, to test the feasibility and acceptability of the intervention, to test trial procedures and to refine the intervention for evaluation in a trial (https://www.journalslibrary.nihr.ac.uk/programmes/hsdr/1264124/#/).

## DEVELOPMENT (STAGE ONE)
### Methods
Stage one of the MRC framework includes three key tasks: identifying the evidence base for an intervention, developing a theoretical understanding of the likely processes

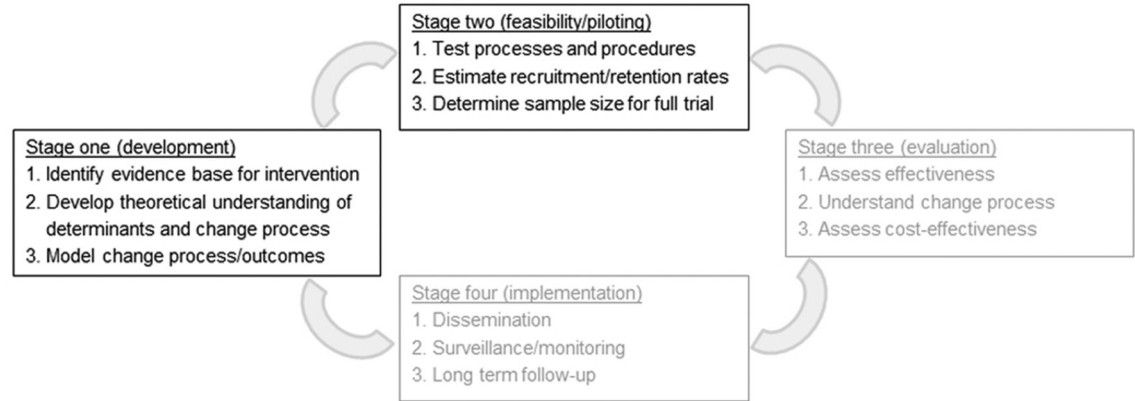

**Figure 1** Use of Medical Research Council framework for developing and evaluating complex interventions.

of change and modelling process to progressively refine the intervention. We reviewed the evidence base on critical care-related psychological risk factors, outcomes and interventions.[1 30] Two authors conducted a new systematic review of psychological interventions in critical care.[25] We also conducted primary research studies[7 31–33] to develop an understanding of the determinants of psychological distress in critical care and the likely mechanisms of change involved in reducing stress (see online supplementary 1).

The modelling process involved testing and revising of the complex intervention, by psychologists, nurses and patients, supervised by an expert psychology advisory group (EPAG) including clinicians, an educationalist and specialists in psychological interventions for psychosis and trauma. This resulted in the nurse-led POPPI (provision of psychological support to people in intensive care) intervention, comprising three interconnected elements: creating a therapeutic environment for all patients, delivering stress support sessions for patients with acute stress and provision of a relaxation and recovery programme. Acute stress was defined by scores≥7 (0–20) on the Intensive care Psychological Assessment Tool (IPAT) previously validated by our group.[34] An education package was created to train staff to deliver the intervention, with help from the EPAG, educational and web designers and a medical film-maker.

### Patient and public involvement

A patients and family advisory group, diverse in age, gender, culture and critical care experiences, was set up at the start of the intervention development phase at University College Hospital, London. Twenty members of the patient group contributed to the early conception of the POPPI study and then to every stage of intervention development. Three patients were members of the EPAG. Five were filmed narrating their critical care experiences, for use in the nurse training courses and to be viewed by current patients as part of the relaxation and recovery programme. The whole group were sent POPPI patient materials as they were devised, for comment and revision.

### Results

Our review of the evidence base[1 30] pointed to likely prevalence rates of 40% clinical anxiety symptoms, 30% depression and 20% post-traumatic stress among survivors. There was increasing evidence that acute stress, as well as early intrusive or delusional memories, were important risk factors for psychological morbidity.[7 20 21] Twenty-three studies were identified in our systematic review of non-pharmacological interventions to reduce critical-care related stress,[35] with 12 showing beneficial effect, although quality and quantity of evidence were limited. However, music, relaxation, visualisation and psychotherapeutic approaches showed promise as elements of a complex psychological intervention.

Our primary research to further understanding of the determinants and proposed mechanisms of change

involved consulting hundreds of patients, relatives and nurses using questionnaires, interviews or focus groups.[7 31–33] Patients expressed the need for understanding and support to deal with anxiety, panic, hallucinations, delusions and flashbacks, as well as the frightening environment in the critical care unit. They wanted to trust staff, feel safe, be listened to and get more information (eg, how drugs might cause hallucinations). They said nurses' ability to communicate with frightened or delirious patients was variable but understood that caring for stressed, delirious patients was difficult and suggested that training could improve communication and empower nurses to deliver one-to-one psychological support. Nurses said they were motivated to improve patients' psychological well-being and make the environment less stressful, but many thought they lacked the necessary knowledge and skills and were frightened of 'making things worse'. They agreed with patients that training would help to increase staff psychological awareness and skills, along with a protocol to help staff identify patients most in need and to deliver one-to-one psychological support. The conclusion of the consultation was that all staff should be trained to reduce general stress and improve communication in critical care, but the most stressed patients would benefit from focused one-to-one psychological support.

As well as adapting techniques from psychological interventions for psychosis and trauma for the needs of critical care patients,[25–28] we drew on well-established work on stress, health and coping,[36] as well as ideas to inform training health staff in new behaviours,[37] including therapeutic communication and engagement with patients, especially those who were distressed by ongoing hallucinations and/or delusions.[38 39] We took into account the need for clinical supervision of non-expert staff delivering psychological support.[40] Finally, we standardised the three interconnected elements of the complex intervention (see table 1) and, guided by the EPAG educationalist, created an education package with associated materials to deliver it. The resulting POPPI intervention is described below. For more details of each element, see online supplementary 2.

In each unit, three POPPI nurses were charged with ensuring that the IPAT was used routinely, encouraging staff behaviour change and overseeing delivery of the whole intervention. The first element of the intervention was for critical care staff to create a more therapeutic environment in their unit by improving communication with distressed patients and reducing stressors such as noise, unnatural light, insomnia, pain and psychoactive drug effects. This was primarily delivered via online training (see below for course details). The POPPI nurse role included encouraging all staff to complete online training and reinforcing key learning objectives through ongoing education and bedside teaching.

The second element consisted of three 30 minute stress support sessions delivered by POPPI nurses to acutely stressed patients (scores ≥7 on the IPAT), ideally starting

**Table 1** The three elements of the POPPI intervention to reduce stress in critical care patients

| Patient interventions | Who receives? | Where? | Proposed mechanisms of change | Training methods | Who is trained? |
|---|---|---|---|---|---|
| Element one: creating a therapeutic environment in critical care | All critical care patients. | In the critical care unit. | Raise staff awareness of patients' stress and psychological morbidity. Motivate staff to reduce stressors in the unit. Train staff in communication skills. Emphasise positive recovery messages. | An online training course. Promotional materials and local education sessions to reinforce key messages. | All clinical critical care staff. |
| Element two: three stress support sessions for patients screened as acutely stressed | Patients screened by Intensive Care Psychological Assessment Tool[22] as acutely stressed (scores 7, range 0–20). | In the critical care unit, on wards following discharge from critical care. | Help patients to express fears or concerns, and process traumatic memories (S1/2). Reduce catastrophic misinterpretations by (A) normalising stressful thoughts and feelings, and (B) testing the reality of fears or concerns (S1/2). Promote hope and motivation for recovery (S3). | A 3-day face-to-face training course for 'POPPI' nurses to learn to deliver stress support sessions and coach patients in using the relaxation and recovery programme. Provision of educational materials (precourse theory booklet, manual and training folder) for POPPI nurses, and ongoing debriefing and support (clinical supervision) from trainers. | Three 'POPPI' nurses per critical care unit, selected by units with reference to suitability criteria. |
| Element three: relaxation and recovery programme on app, DVD and booklet (music, relaxation, meditation, patient recovery videos and self-help information). | Patients screened as acutely stressed and receiving stress support sessions. | In the critical care unit (via tablet computer) and at home (via DVD and self-help booklet). | Provide distraction and meaningful activity. Reduce stress by teaching patients relaxation and coping strategies (in S1/2). Normalise emotional reactions and promote optimism by providing former patients' recovery videos (in S1/2). Promote recovery by providing information and helping to create a personal plan for future well-being (DVD and booklet; S3). | | |

S1, session one; S2, session two; S3, session three.

the sessions in the critical care unit within 48 hours of IPAT screening and completing the three within a week (in the unit or on other wards). The three stress support sessions were based on psychological techniques evaluated in other populations and adapted for critical care patients. Techniques from psychological interventions for psychosis included establishing a collaborative relationship focused on reducing distress; managing patient concerns based on hallucinations and delusions; psychological education to reduce distressing interpretations of unusual experiences, reduce stigma and encourage open communication; and provision of active coping strategies. Interventions were adapted to take account of critical care patients' likely physical, emotional or cognitive fragility.

Stress support sessions also included psychological techniques from trauma-related interventions to help people to identify unhelpful thoughts, interpretations and coping styles and to think about their experiences in a less upsetting way. However, some psychological interventions for trauma are not suitable for critical care patients. For example, during stress support sessions, patients were encouraged to express their thoughts and feelings about their critical care experiences if they wished, but not to undertake 'reliving' their critical care stay, particularly as their trauma may be continuing.

The main objectives of the stress support sessions were for nurses to develop a trusting relationship with patients, so patients could discuss concerns that they might feel embarrassed or worried about communicating, and to reduce emotional distress. Stress support session one involved discussing and normalising common psychological reactions in critical care, helping patients to open up about worries and exploring coping strategies. In the second stress support session, patients were encouraged to open up more fully about their fears, to identify highly stressful thoughts and to learn how to find out if their worst fears were realistic. The third session began with summarising key messages and reviewing remaining problems. Nurses then helped patients to make a personal plan to cope with challenges ahead, to build realistic optimism about recovery and plan where to seek help if problems arose in future. Patients were asked to rate their stress levels on a 'stress thermometer' (a simple tool commonly used by psychotherapists to rate stress levels during therapy on a scale of 0–10) at the beginning and end of each stress support session, to help nurses monitor how patients were feeling as the sessions progressed.

The third element of the complex intervention was a relaxation and recovery programme designed to help people practise new coping strategies, to provide meaningful activity and distraction and to learn from other patients' experiences, between stress support sessions. The POPPI app with calming music, relaxation, meditation, nature sounds/videos and patient recovery stories was loaded onto tablet computers loaned to patients receiving stress support sessions. DVDs containing similar materials to the app and a self-help booklet 'Getting well, staying well' were given to patients to keep and take home.

An education package was developed to train staff in each element of the intervention. We adopted a 'blended learning' approach, combining online training and face-to-face teaching. Interactive and engaging online training courses are recommended by educationalists for the provision of knowledge to a large number of staff, while face-to-face training is suitable for adults acquiring new skills by practising and receiving feedback.[41] We evaluated training by measuring nurses' reaction to the courses, learning and self-confidence in psychological skills.[42] The online training course to deliver element one had four aims: to increase knowledge and understanding of the psychological impact of critical care; to learn to reduce stressors in the environment; to improve staff communication skills; and to promote realistic hope for patients' recovery. It was co-designed with education experts as an interactive course using videos, colourful graphics and quizzes. On completing the course, staff took an online knowledge test and received certificates if they passed.

Additionally a central face-to-face course was devised to train POPPI nurses to deliver all elements of the complex intervention. During the three-day course, psychological principles were taught, and all three elements of the intervention were covered. A significant amount of time was devoted to skills practice in delivering stress support sessions, with the training team (a psychologist, nurses and patient representatives) observing and offering feedback. An additional day of feedback and assessment was held after POPPI nurses returned to their units and delivered sessions to at least one patient each. The day included nurse focus groups on their experience of delivering stress support sessions, and a competence assessment, in which nurses delivered stress support session two to a simulated patient (actor), observed by trainers who completed a checklist. Thereafter, nurses had clinical supervision with trainers during regular 'debriefing and support' phone calls.

Associated training materials were developed, including an intervention manual for POPPI nurses, a set of slides for the three-day training course and a training folder consisting of course handouts, patient scenarios, stress thermometers, summaries and checklists of each stress support session and reflective note templates.

Key psychological outcomes were identified as PTSD and depression, as the most serious common psychological outcomes of critical care. Research in identifying modifiable critical care risk factors is more extensive for PTSD than for depression and anxiety; therefore, PTSD was chosen as the primary outcome of the full trial.

## FEASIBILITY AND PILOTING (STAGE TWO)
### Methods
An intervention feasibility study was held in two English adult general critical care units (UCLH and Watford General Hospital), and a different trial procedures feasibility study was held in two other hospitals (Bristol Royal Infirmary and Medway Maritime Hospital).

### Intervention feasibility study

The aims of this study were to test the feasibility and acceptability of content and delivery of the complex intervention and to get feedback from patients and staff to help refine the intervention prior to full evaluation. No follow-up questionnaires were sent to patients in this study. Each hospital was asked to send three 'POPPI' nurses to attend the central three-day face-to-face course to learn about content and delivery of the intervention. Once they returned to their hospitals, all clinical staff in their units were asked to complete the online training course on creating a therapeutic environment, and patient recruitment began. POPPI nurses also attended an additional day of feedback and assessment once each had conducted stress support sessions with at least one patient.

Inclusion criteria for patients were: age 18 years or greater; receipt of level 2 or 3 care for 48 hours or more; Richmond Agitation Sedation Scale[43] score between +1 and −1; ability to speak English and ability to communicate orally; no pre-existing cognitive impairment, psychotic illnesses or chronic PTSD; Glasgow Coma Scale score of 15 and not terminally ill or receiving end-of-life care. Patients were provided with written and verbal information as part of the informed consent process. Consenting patients were screened for acute stress (≥7 on the IPAT). Stress support sessions and a relaxation and recovery programme were delivered to acutely stressed patients by POPPI nurses.

Feasibility and acceptability of content and delivery of the intervention were assessed using quantitative and qualitative data from nurses and patients (see table 2). Quantitative data included delivery rates of each element of the intervention, as well as scores/ratings from patient satisfaction questionnaires and staff training evaluation questionnaires (on satisfaction, learning and self-efficacy). Staff satisfaction and learning questionnaires were completed at the end of the online training course and the three-day face-to-face course. POPPI nurses also completed self-efficacy (confidence in psychological skills) questionnaires before and after the three-day course and on the additional feedback and assessment day, when a competence assessment was also carried out with a trainer for each nurse. Qualitative nurse data were collated from focus groups on the additional training day, and at the end of the study, nurse debriefing sessions with trainers and from case report forms (CRFs). Once the intervention was embedded in the intervention sites, patients who received the stress support sessions and relaxation and recovery programme completed satisfaction questionnaires. Qualitative patient data came from free-text sections on the satisfaction questionnaires and nurse-reported patient comments in the CRF. The above data were used to refine the intervention, where relevant.

### Trial procedures feasibility study

The aim of this study was to determine recruitment and retention rates for a full trial. Patient eligibility criteria were the same as for the intervention feasibility study, and

consent was obtained. The intervention was not delivered in this study. At 5 months postrecruitment, patients were sent follow-up psychological and health questionnaires proposed for the full trial. These included the PTSD Symptom Severity Scale – Self Report (PSS-SR)[44] and the 10-item Centre for Epidemiologic Studies Depression Scale (CES-D)).[45]

### Patient and public involvement

The patient group continued their involvement with POPPI throughout the feasibility phase. Patients attended the nurse 3-day training course, both to tell their own critical care stories and to observe and give feedback during nurses' skills practice to deliver the stress support sessions. They helped to interpret results of the feasibility study and would go on to play a key role in the full POPPI trial. Additionally, patient participants in the studies were given a link to the POPPI website so they would be able to read results of the studies.

## Results

### Intervention feasibility study

#### *Delivery of the intervention*

Ten POPPI nurses (100%) from the two hospitals completed a central three-day face-to-face training course (due to personnel issues, this included training extra nurses at one site). When they returned to their hospitals, online training became available for all staff, and patient recruitment was opened for 5.5 months. Two hundred and eighty-three (84%) staff in the two critical care units took the online training course, surpassing the target rate of 80%, with 277 (98%) passing the end of course assessment. One hundred and twenty-seven patients were recruited to the study, all were screened with the IPAT and 51 (40%) were identified as acutely stressed (see figure 2). Forty-four stressed patients (86%) received stress support sessions, with 39 (77%) receiving two or three sessions and five having one session only. Seven patients (14%) had no sessions because they declined, were discharged home early or deteriorated clinically. Median duration of sessions one to three was 35, 30 and 30 min, respectively. Forty patients (90%) received a tablet computer, and 27 (61%) were given a DVD and patient booklet to keep.

#### *Content of the intervention*

All nurse and patient feasibility questionnaires had scales of 0–5 per item, with 4 or 5 defined by us as 'good' scores or ratings. Staff gave the online training course good ratings for: stimulating (73%, 189/260); useful (86%, 223/260) and well designed (84%, 216/257). All 10 POPPI nurses rated the face-to-face course as good (stimulating, useful, well-conducted and motivating), with nine (90%) rating relevance to their new role as good. There was a large increase in nurse self-efficacy, from 30% good scores (21/70 across seven items) precourse to 73% good scores (51/70) postcourse. The same proportion of 73% good scores was maintained at follow-up. For nurse

**Table 2** Feasibility, acceptability and refining of the three elements of the psychological intervention

| Elements of intervention | Content/delivery of element | Feasibility and acceptability indicators – quantitative and qualitative | Feasibility and acceptability results | Refinement of the intervention postfeasibility study |
|---|---|---|---|---|
| Element one: creating a therapeutic environment in critical care | Content of online training course. | Training course ratings by all staff (% with 4 or 5 (0–5) or 'good'). | Stimulating: 73%; useful: 86%; well-designed: 84%; right length: 80% (n=260, but missing data for some items). | Online training course shortened and made more visually appealing, with more practical advice on reducing stressors in critical care units and clearer presentation of key messages. |
| | | Favourite parts of course – all staff. | Factual information: 42 %; patient stories : 35%; communication videos : 13%; tests 10% (n = 260) | |
| | | Nurse qualitative feedback.* | Staff positive, suggested minor improvements . | |
| | Delivery of online training and creating a therapeutic environment. | Staff taking course (target: 80%). | 283 (84%). | Provision of training, display materials and slide sets for seminars/workshops for local education teams to support and motivate staff in creating a therapeutic environment. |
| | | Staff passing final test (score > 80%) . | 277 (98%). | |
| | | Staff learning scores (% with 4 or 5 (0 – 5) or ' good'). | 74 % (n=259). | |
| | | Nurse qualitative feedback. | POPPI nurses lacked time, due to workload, to support staff in creating therapeutic environment. | |
| Element two: three stress support sessions for patients screened as acutely stressed | Content of screening. | Previously validated. [34] | | |
| | Delivery of screening. | Consenting patients screened. | 127 (100%). | |
| | | Screened as acutely stressed. | 51 (40%). | |
| | Content of stress support sessions. | Median (IQR) difference in patient stress thermometer scores (0–10). | Median difference from start session 1 to end session 3 was –3.0 (–5.0 to –1.0) (n=25 patients who had all three sessions). | Content of stress support sessions clarified for POPPI nurses and patients by reorganising sessions from five components each into three common components in all sessions and three individual components per session. Manual became more tightly focused on stress support sessions (rather than the whole intervention) with clearer signposting to and between sections. |
| | | Patient satisfaction with stress support sessions (% with 4 or 5 (0–5) or 'good') . | Overall: 93 % ; h elped express fears : 93 %; n urse understanding : 100 %; n urse normalised fears : 100 %; f ewer stressful thoughts : 87 % ; fewer stressful feelings : 80 %; number/duration of sessions : 80 % (n=15 , missing data some items). | |
| | | Patient qualitative feedback . | Stress support from nurses was very helpful . | |
| | | Nurse qualitative feedback . | Rewarding but challenging to explain to patients. | |
| | Delivery of stress support sessions. | Number of stress support sessions patients had. | 25 (49%) had three sessions; 14 (28%) had two sessions; 5 (10%) one session; 7 (14%) had none. | Ensure buy-in and support for POPPI nurses from clinical, education and research staff from the start, making it a team effort. If hospital discharge is near, sessions 2 and 3 can and should be delivered together. |
| | | Median duration of sessions. | Session 1 : 35 min; s ession 2: 30 min; s ession 3: 30 min. | |
| | | Nurse qualitative feedback. | Nurses struggled to find time in their daily work schedule to conduct stress support sessions , especially if patients postponed. Patients missed session 3 if they were discharged home early . | |

Continued

**Table 2** Continued

| Elements of intervention | Content/delivery of element | Feasibility and acceptability indicators – quantitative and qualitative | Feasibility and acceptability results | Refinement of the intervention postfeasibility study |
|---|---|---|---|---|
| | Content of POPPI nurse face-to-face training course – 3 days and a feedback/assessment day. | Nurse feedback – postcourse questionnaire (% with 4 or 5 (0–5) or 'good'). | Stimulating: 100%; useful: 100%; relevant: 90%. Well conducted: 100%; motivating: 100% (n=10). | To reduce burden and increase self-efficacy, the 3-day course became more focused on stress support sessions, particularly session 2, seen as the most difficult session to deliver. More emphasis on skills practice, with actors (not fellow trainees) playing patients. Wider spectrum of patient scenarios used. Assessment to be reframed as 'skills development' and carried out one-to-one, with a trainer playing the patient. |
| | | Nurse self-efficacy (in delivering psychological support) – precourse and post-course and feedback day questionnaires. (% of 4 or 5 (0–5) or 'good' ratings of items. | Big increase in self-efficacy from precourse to post-course; maintained at follow-up. 30% of scores (21/70) across items were good pre-course; 73% were good scores (51/70) post-course and on feedback day. | |
| | | Nurse course learning – an eight-item post course knowledge questionnaire (% of 4 or 5 (0–5) or ' good' scores for items) | 87% of scores (69/79) were 'good' for learning on: acute stress; screening; aims of stress support sessions, normalising, psycho-education, communication style, stressful thinking, checking out fears and, coping. | |
| | | Trainer assessment of nurse competence using six-item checklist (scores 0–12; pass=6). | 100% passed (9 on first attempt, 1 on second). Median (IQR) score of passes 9 (9, 10). | |
| | | Nurse qualitative feedback. | Course highly valued but tiring. Skills practice stressful. Competence assessment on follow-up day stressful. | |
| | Delivery of 3-day training course. | Number of required trainees attending. | 10 (100%).† | Some modules dropped from course; precourse booklet on psychological principles was provided. |
| | Content: debriefing support by trainers. | Nurse qualitative feedback. | Debriefing calls useful for reflection, confidence. Assessment should be part of ongoing support. | Assessment to be one-to-one confirmation of skills (as above), as part of ongoing debriefing and support. |
| | Delivery: debriefing and support. | Nurse qualitative feedback. | Nurse debriefing and support should start earlier. | First debriefing call after first POPPI patient. |
| Element three: relaxation and recovery programme on app, DVD and booklet (music, relaxation, meditation, patient recovery videos and self-help information) | Content of relaxation and recovery programme. | Patient satisfaction with content of programme (% with 4 or 5 (0–5) or 'good'). | Content on tablet computer app: 71%; useful post-ICU coping ideas: 67% (15 patients, some missing data). | Content and design of the relaxation app were improved. Balance of contents of DVD were improved, and calming classical music tracks were added. Layout and readability of the patient booklet were improved. |
| | | Nurse-reported qualitative patient feedback. | Varied preferences; relaxation, meditation, nature sounds, patient stories or calming music. Some were disappointed not to find calming classical music on DVD. | |
| | Delivery of relaxation and recovery programme. | Patients receiving tablet in session 1. | 40 (90%). | Usability of the relaxation app was improved to make it easier for the less dextrous. Higher spec tablets, including better touch sensitivity, identified for use in future trial. DVD and booklet to be provided in session 2, so more patients would receive them (many missed receiving session 3 due to being sick or discharged from hospital). |
| | | Patients receiving DVD or booklet. | 27 (61%). | |
| | | Nurse-reported qualitative patient feedback. | While some liked tablet, others found it hard to use; some preferred DVD or patient booklet. | |

*All nurse and patient qualitative data described on pp8-9.
†Four additional nurses were trained at one site due to personnel issues.
ICU, intensive care unit.

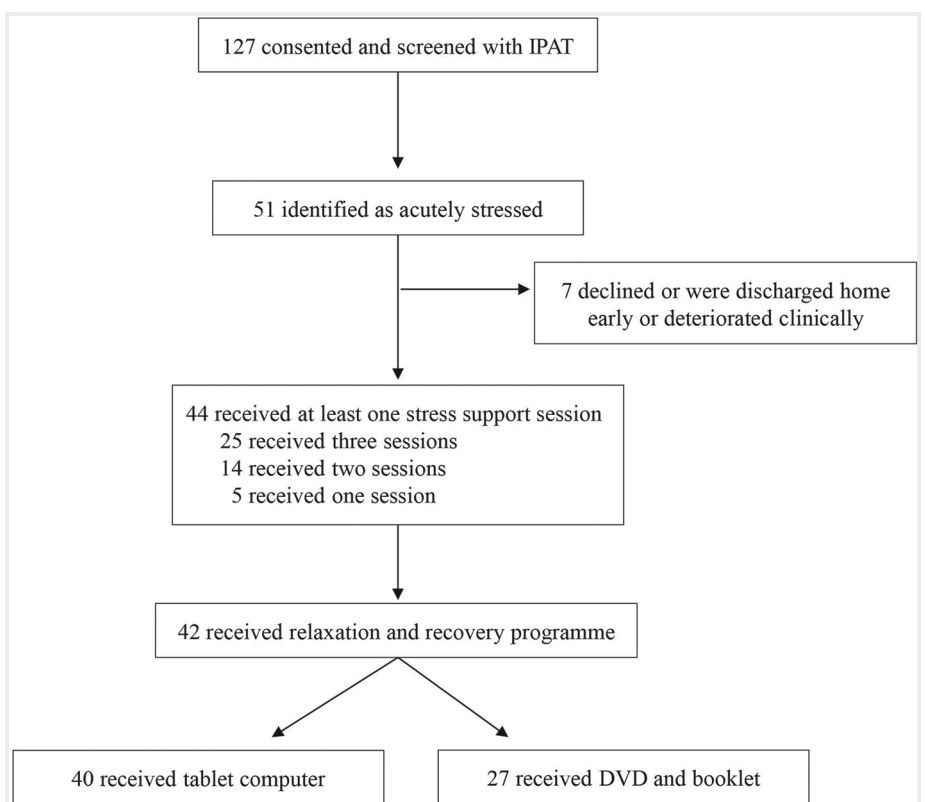

**Figure 2** Patient flow in the intervention feasibility study. IPAT, Intensive care Psychological Assessment Tool.

learning, 87% (69/79) of postcourse scores on eight items were good.

Once the intervention was embedded, 15 patients completed a satisfaction questionnaire. Satisfaction with stress support sessions was rated as good by 14 (93%) patients. Fifteen patients (100%) gave good ratings for nurses understanding them and normalising their fears. Slightly fewer patients thought their stressful thoughts (13, 87%) or feelings (12, 80%) were reduced by the sessions. Twelve (80%) rated the number and duration of sessions as 'just right'. Satisfaction with the relaxation and recovery programme was somewhat less than for the stress support sessions. Ten of 14 (71%) rated contents of the tablet app as good, while 10 of 15 (67%) thought that they learnt good coping strategies from the programme. Stress, rated on the stress thermometer (0–10), was reduced by median three points (IQR 1–5 points) from the start of session one to the end of session three among the 25 patients who received all three sessions.

Quantitative feasibility and acceptability data are presented fully in table 2. Qualitative data on the three elements are also summarised in the table, but specifics, both positive and negative, are described in more detail as follows. Nurses reported that the online training course had raised awareness, changed staff thinking and led to a better environment in their units. However, they said design improvements, more practical advice and summaries of key messages were needed. Delivery of element one was challenging, as POPPI nurses were too busy to support other staff in creating a therapeutic environment.

It was suggested this responsibility should be shared by the wider critical care team, and further educational/promotional materials were needed.

Nurse feedback on content of the second element was that stress support sessions were welcomed by patients and rewarding for nurses. Challenges included ensuring patients understood the sessions and knowing how to handle very serious issues that might be raised. Delivery of stress support sessions could be difficult to fit into heavy clinical workloads, especially if patients postponed sessions due to fatigue or sickness.

Comments on the three–day course included positive ones such as 'best course ever' and 'feel privileged' to be part of it. However, some nurses said they found it hard work and tiring. They suggested it would be less intense if psychological theory was delivered to be studied in advance, rather than presented during the course, and that skills practice should involve actors (instead of fellow trainees) to play patients. They found the POPPI nurse training manual invaluable but suggested some restructuring and clarification in the presentation of stress support sessions, with better linkage to appendices. All nurses found the competence assessment on the additional training day stressful. Most said they welcomed the regular debriefing and support calls with trainers, finding them useful for reflection and building greater confidence.

Patient feedback on the stress support sessions was mainly positive and included 'every hospital should have it', 'it seemed a life saver', 'proves that I am normal

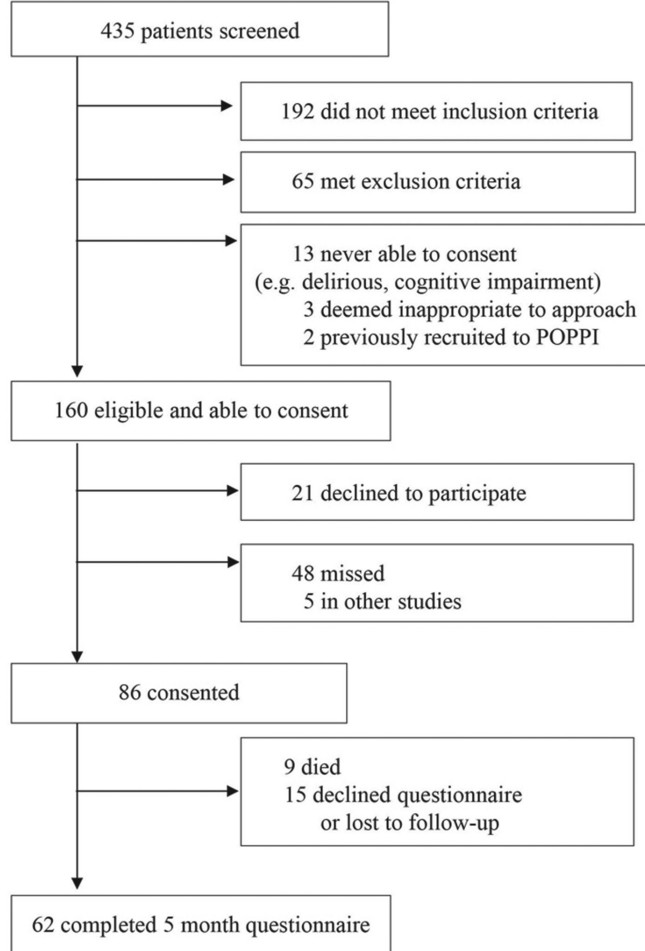

**Figure 3** CONSORT diagram for the trial procedures feasibility study. CONSORT, Consolidated Standards of Reporting Trials.

but under the influence of morphine' and 'I hope that POPPI will eventually be used in all ICUs. Thank you for the support'. Criticisms included one patient saying that the critical care environment had not been improved in response to his concerns, and some patients wanted more than three stress support sessions. Patients also responded well to the relaxation and recovery programme. Some patients preferred patient stories, some relaxation exercises, some meditation and some calming music. However, some found the tablet computers confusing and difficult to use, due to unfamiliarity with the technology, or reduced dexterity. Unfortunately, the tablets used in the intervention feasibility study did not have good touch sensitivity. Some patients were disappointed that the DVDs did not include the calming music (due to copyright issues).

### Refinement of the complex intervention
We refined the complex intervention based on key points emerging from the feasibility data (see table 2, last column). To refine element one, we made the online training course shorter, more attractive and clearer and included more practical advice and emphasis on key messages. We created new slide sets and display materials for units to reinforce the key messages to their staff. We decided that the wider critical care team should be involved in creating the therapeutic environment, so that POPPI nurses could focus on delivering stress support sessions to patients.

To improve element two, the three-day face-to-face POPPI nurse course was more tightly focused on the stress support sessions, with more emphasis on skills practice, and actors playing patients, to reduce the burden on trainees. Some modules, such as a booklet on psychological principles, were provided to POPPI nurses in advance to reduce the intensity of the course. The nurse competence assessment was reframed as a skills development assessment and carried out less formally as part of ongoing one-to-one debriefing and support sessions with a trainer. The manual was made simpler and clearer to improve nurse and patient understanding of the components of the stress support sessions. As many patients in the feasibility study did not receive the third support session due to early hospital discharge, it was decided that sessions two and three could and should be delivered together in those circumstances, if patients agreed and were well enough.

To refine element three, the content, design and usability of the relaxation app was improved, and tablets with better touch sensitivity were identified for use in the future trial. Small improvements were made to the design and content of the DVD and patient booklet, and it was decided they would be given to patients in session two rather than session three, so that more patients would receive that element of the intervention, even if they were due to leave hospital after session= two.

### Trial procedures feasibility (study two)
There was a two-month recruitment period. Of 160 eligible patients in the two sites, 86 (53.8%) provided consent to take part (see figure 3). Of these, nine (10.5%) had died by five months, but 62 (80.5%) completed the five-month follow-up questionnaire. Overall completeness of the primary outcome measure (the PSS-SR[44])was very good. For a total of 1054 fields, only 24 (2.3%) had missing data.

### DISCUSSION
The objectives of the POPPI intervention development study were met: we developed and standardised a nurse-led psychological intervention to reduce patients' acute stress in critical care; we created a staff package to deliver it; and we tested feasibility of the intervention and trial procedures and refined the intervention. The intervention had three interconnected elements: creating a therapeutic environment; stress support sessions delivered by selected, trained nurses to acutely stressed patients; and a relaxation and recovery programme for acutely stressed patients. To support delivery of the intervention, we devised and created: a relaxation and recovery app, DVD and booklet for patients; an online training course

for all critical care staff; and a training package for POPPI nurses, including a manual of the three stress support sessions, a three–day face-to-face course with training folder and a debriefing and support programme.

An intervention feasibility study showed that delivery and content of the complex intervention were feasible and acceptable for patients and staff and led to a refined version of the intervention, ready for evaluation in a cRCT. A separate trial procedures feasibility study demonstrated that the necessary recruitment and retention rates for a cRCT could be achieved.

## Strengths of this study

The intervention builds on extensive prior work investigating the psychological impact of critical care, and it was developed using a rigorous framework[29] with input from a large patient advisory group and supervision from an expert team. The detailed development work resulted in a novel nurse-led preventative psychological intervention to improve psychological outcomes. Importantly, the intervention has three interconnected elements designed to: reduce stress in the environment and improve staff-patient communication; provide targeted support for the most stressed patients; and provide stress relief and meaningful activity for patients. Feasibility of both delivery of the intervention and trial procedures were thoroughly tested in two studies, involving more than 200 patients and 280 staff. The psychological intervention was confirmed to be feasible and acceptable for patients and staff. Formal evaluation of the intervention in the POPPI cRCT is underway.

Limitations were that the intervention was piloted in only two sites and limited patient feedback was gathered. No efficacy data were collected during the feasibility studies.

The POPPI intervention addresses evidence that critical care patients suffer acute and long-term stress.[7–10] Despite increasing recognition[13] that interventions to reduce critical care stress are needed, little is known about what would be most effective. Studies of early interventions such as music, relaxation techniques and psychological support in critical care showed promise.[23 24] POPPI is the first complex psychological intervention designed to be delivered early, during the critical care admission, to prevent PTSD and other long-term morbidity.

## CONCLUSION

The POPPI psychological intervention to reduce stress and prevent long-term morbidity for critical care patients was feasible, acceptable and ready for evaluation in a cRCT.

## Author affiliations
[1]Critical Care Department, University College London Hospitals NHS Foundation Trust, London, UK
[2]Psychological Interventions Clinic for Outpatients with Psychosis, Maudsley Psychology Centre, Maudsley Hospital, London, UK
[3]Division of Psychiatry, University College London, London
[4]Research Department of Clinical, Educational & Health Psychology, University College London, London, UK
[5]Institute of Pharmaceutical Science, King's College London, London, UK
[6]Intensive Care National Audit and Research Centre, London, UK
[7]University College London/ University College London Hospitals NIHR Biomedical Research Centre, Institute of Sport Exercise and Health (ISEH), London, UK

**Acknowledgements** Daniel Freeman (chair of the education and psychology advisory group), Dane Goodsman (clinical educationalist), Rob Beer (online training designer), Daniel Zanitchkhah (app designer), Derek Tutssel (video director), Lih-Mei Liao, Wendy Harris, Steve Saunders, Ruth Canter, Rahi Jahan, Blair McLennan and Emma Wood. David Aaronovitch, Sheila Richards, Peter Cross, Rajadurai Sunderalingham, Kate Baden-Fuller, Caroline Knight and other members of the UCLH patients and family advisory group. 'POPPI' nurses and site PIs: University College Hospital, London: Emma Davies, Emma Thompsett, Rosalind Edwards, David Brealey, Magda Rocha, Jung Ryu and Georgia Bercades. Watford General Hospital: XioBei Zhao, Jennie Haydock, Kalpana Giri Ghimire, Lillian Norris, James Cunningham and Sarah-Jane Turner and Tracey Temple. Medway Maritime Hospital: Catherine Plowright, Claire Pegg, James Cullinane. Bristol Royal Infirmary: Katie Sweet, Sanjoy Shah and Lisa Grimmer.

**Contributors** Early conceptual and development work for the POPPI intervention was carried out by DW, JWei, DH, MM, DS, JWel, NA, MH, CW and CB at University College London Hospitals. Further development of the intervention and design of the feasibility phase of the POPPI study were carried out with KMR, SH, DAH, PRM, ARB, DD and VB. DW drafted the initial manuscript with input from KMR. All authors revised the manuscript critically and gave their approval of the final version. DW is the guarantor.

**Funding** This project was funded by the National Institute for Health Research Health Services and Delivery Research Programme (Project: 12/64/124).

**Disclaimer** The views and opinions expressed in this paper are those of the authors and do not necessarily reflect those of the HS&DR, NIHR, NHS or the Department of Health.

**Competing interests** DW also received funding from the University College London/University College London Hospitals NIHR Biomedical Research Centre.

**Patient consent** Not required.

**Ethics approval** The two feasibility studies were granted favourable ethical opinion by the NRES Committee South Central – Oxford B research ethics committee on 23 April 2014 (reference: 14/SC/0149), were managed by the ICNARC CTU and prospectively registered on the ISRCTN Registry (ISRCTN61088114).

**Provenance and peer review** Not commissioned; externally peer reviewed.

**Data sharing statement** No additional data available.

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
