## [Reviewer comments · BMJ Open]

ARTICLE DETAILS

TITLE (PROVISIONAL)	Providing psychological support to people in critical care: development and feasibility study of a nurse-led intervention to prevent acute stress and long-term morbidity
AUTHORS	Wade, Dorothy; Als, Nicole; Bell, Vaughan; Brewin, Chris; D'Antoni, Donatella; Harrison, David; Harvey, Mags; Harvey, Sheila; Howell, David; Mouncey, Paul; Mythen, Monty; Richards-Belle, Alvin; Smyth, Deborah; Weinman, John; Welch, John; Whitman, Chris; Rowan, Kathryn

VERSION 1 – REVIEW

REVIEWER	Julie Darbyshire University of Oxford, UK
REVIEW RETURNED	01-Jan-2018

GENERAL COMMENTS	This was a very clear and well-written manuscript that outlines the work needed ahead of time for a randomised controlled trial. The area of focus is of very high importance to patients and their families (see James Lind Alliance top ten for critical care) and it seems likely that early support could help prevent long-term psychological uncertainty setting in before ICU discharge. The good, robust RCT that will follow this feasibility programme is awaited with interest... I have very minor comments only on the manuscript submitted for review. I was surprised that the author list doesn't include an educationalist when the development of the training package was a significant part of the feasibility work, and the refinements of the intervention have clearly been made in line with good adult educational theory. Linked to this thought, there is no mention of educationalist support in the development section (page 6, line 33) although this is referred to later (in the acknowledgements). As a style point, I wasn't sure that the early stage doctoral/post-doctoral work needed to be highlighted in quite the detail that it is. "Prior work" with refs seems fine, and would be clear enough for the reader to understand that the POPPI feasibility work builds on earlier work by members of the team. Page 7, line 9: for clarity I suggest including the IPAT score for "acutely stressed" Page 7, line 40: it's not entirely clear if the patient films are of experiences of ICU or experiences of PPI as part of the study programme (I assume the former!)
---

	Page 9, line 52: It would be useful to include the timeframe for the three sessions to be included in the main body of the text as well as in the appendix Page 12 lines 4-26: Were there any negative comments/scores in the feedback? These would be worth exploring (a bit) because non-mainstream thoughts can indicate some interesting points for future consideration.
--	---

REVIEWER	R Hopkins Intermountain Medical Center, USA
REVIEW RETURNED	03-Jan-2018

GENERAL COMMENTS	Introduction. Page 4 line 9 "long-term distress"; should be "long-term psychological morbidities"; Page 4 line 28 "psychological support in their recovery plan"; should be changed to "psychological support as part of a recovery plan"; Second paragraph of the introduction - there is a growing literature of interventions to improve psychological outcomes including seminal work by Christina Jones on ICU diaries and the work by Chris Cox on mindfulness and resilience. These studies should be cited. It is unclear what outpatient services (third paragraph page 4) have been evaluated - see comment above. Developmental stage one - please describe why the MRC framework was chosen. Not all readers outside the UK will be familiar with it. Briefly describe the intervention in third paragraph on page 6. Also describe the expert psychological advisory group. The two feasibility studies were run simultaneously it is unclear what the difference between the two studies are - the paper says one was an intervention feasibility study and one was a trail procedure study to obtain recruitment and retention rates - this data was collected in both studies collected patient recruitment information. I am sure I am missing an important point here, but several readings has not improved my clarity on this point. It would improve the clarity of the paper if the development of the intervention and the results were discussed before the feasibility studies. Since this is not a true experimental design, this should be easy to do. This organizational change would reduce the memory load by discussing all of development and then move to the feasibility studies. Methods of the feasibility studies state the delivery and content of the intervention were assessed - please state how assessed, questionnaire to patients and POPPI nurses? Patient and public involvement - suggest changing to the Patient and Public Advisory Group - methods should describe who is in the group and how they were selected. The manuscript states 6 were in the training team, not sure what this means, but helped to train the POPPI nurses? Was the group used in only the feasibility studies or in both the development and feasibility stages?
--

	The first sentence of the results could be deleted. Rather start with a brief introduction and then step one. page 8 first paragraph - the literature review found 12 studies with beneficial effects it would help to summarize what was beneficial. The paper discussed theoretical frame works and a developmental model that were used but there is no description of the theories or rational as to why they were picked. While behavior change is certainly needed, the paragraph on the COM-B model of behavior change seems separate from the development of the intervention - suggest deleting this section and for clarity only focus on development (and remove from table 2). An entire paper could be written on the behavioral change aspects of development and implementation of the intervention. Table 2 - column theoretical approach - While important, there is little discussion of this in the paper or table (I assume this was detailed in the dissertation) but I wonder if the paper and table should focus on the intervention and less the models. This would help shorten and focus the paper. If the models are kept in the paper, a description of the models and a link to the intervention needs more description. It is unclear if the IPAT and the stress thermometer are the same measurement tool? Why do only patients who meet a stress criteria receive the stress support sessions? is the IPAT used after to see if the intervention is effective? Same questions for element 3 of the intervention. page 10 first paragraph. Please briefly describe the cognitive and behavioral interventions. Page 10 last paragraph, please describe the Kirkpatrick model of evaluating training. How did it help refine the intervention? Recommend that the feasibility methods and results are reported together. Most of the results of the pilot study are in Table 3. The results should summarize the most important findings emphasizing how the intervention was improved. strengths of the paper are the development of a novel psychological intervention to improve psychological outcomes. (page 16 second to last paragraph) No need to say it was part of dissertation, cite dissertation. Also important are the 3 parts of the intervention, and that it is feasible to use. Data collection under way to evaluate intervention in future studies. The two study diagrams could be simplified into one if the two feasibility studies are combined. see comments above (of course I may be missing an important reason not to do this). Online data supplement is very helpful.
--	---

VERSION 1 – AUTHOR RESPONSE

Reviewer 1.

Thank you for your generous and helpful comments. We will respond to your points in order:

1. We have made further references to the educationalist's support and described her role in the expert advisory group on page 4 para 3 and page 5 para 3.
2. We agree that the doctoral work was over-emphasised and have changed that to prior work/reference throughout the paper. The table detailing the prior work has been moved to the electronic supplement (ES one)
3. We have added the IPAT score for "acutely stressed" in Development (stage one)/Methods p4, para 3 and repeated it in Results (p6, para 2)
4. We have clarified that the patient films are of their critical care experiences (last sentence, p4)
5. Time frame of intervention now clarified in text - please see first sentence, second paragraph, p6
6. We have tried to present more of a balance of positive and negative comments, although in honesty there were not many negative comments/scores. Please see p10 last sentence - p11 first paragraph. In general p11 contains a balance of positive and negative feedback from nurses and patients, which we used to refine the intervention.

Reviewer two

Many thanks for your thorough review and helpful suggestions, that have led to us revising the paper as follows:

1. We have used the term "psychological morbidity" on p3, line 5 and throughout the paper
2. See p3, para 2, line 2 - amended to "part of a recovery plan"
3. See p3, para 2, refs 18 and 19 – re Christina Jones on patient diaries and Christopher Cox on mindfulness/coping skills training
4. See p3, para 2 for more detail on which post-critical care discharge services have been evaluated to date
5. See p4 para 1 for rationale for using the MRC framework
6. The intervention is now briefly described in Development/methods, see p4, para 3, second sentence.
7. We have tried to clarify the difference between the feasibility studies throughout the paper - they were two separate studies carried out with different participants at different hospitals, with different aims. We're sorry this was not clear enough in the first version, and hope we have done a better job this time round. Please see pp 9-10
8. We changed the organisation/structure of the paper as requested. Thanks for this very useful suggestion.
9. We have clarified the methods of how feasibility was assessed - see expanded paragraph on this on p9, last paragraph
10. See page 4, last para, for this revision. Patient group involved from beginning of project to now
11. First paragraph of Results has been reworked. See page 5, para 1 also for sentence on which interventions were beneficial
12. We have defined what we mean by "theory" more carefully in the paper - ie a theoretical understanding of the process of change" and reduced focus on models, as you suggested. See for example p4 para 3, and p5 para 2 and 3. See table one, page 7.
13. Agreed. Used COM-B as reference and just alluded to behaviour change in text (p5 para 3)
14. Thanks, we have amended table one, p7 accordingly – see col 4 "proposed mechanisms of change"
15. The IPAT and the stress thermometer are not the same tool. See p6 para 4, last sentence for further description of the stress thermometer.
16. We thought that the intervention should be targeted at those who would benefit most. The IPAT was not used afterwards to see if the intervention was effective, as the feasibility studies were not powered to test effectiveness (which is being evaluated in the cRCT) as that was not their aim.

17. Cognitive and behavioural interventions (now described in the paper as psychological interventions, to reduce the emphasis on models) are described on p6 para 2-4, and in electronic supplement two
18. The text no longer talks about Kirkpatrick model, it is referenced only, and the relevant elements are described on p8 lines 6-7.
19. We followed your helpful recommendation to report feasibility methods and results together (p9,10 etc)
20. We have rewritten/reorganised the feasibility results section to summarise the most important findings emphasising how the intervention was improved (see pp 10, 11 and 14)
21. We have rewritten strengths/limitations according to your helpful suggestion. See pp2 and 15
22. We were unable to combine the two study diagrams as the studies used different patients and the aims were different, as I hope we've now clarified (see point 7 above)
23. Thanks, we are pleased you found the supplement helpful.
- Thanks again for your thorough review, which we believe has helped us to strengthen the manuscript

VERSION 2 – REVIEW

REVIEWER	Julie Darbyshire University of Oxford, UK
REVIEW RETURNED	21-Feb-2018

GENERAL COMMENTS	This revised version is much more comprehensive and describes the project more clearly. I have no additional comments, queries, or suggestions.
---

REVIEWER	Ramona Hopkins Intermountain Medical Center, United States of America
REVIEW RETURNED	21-Mar-2018

GENERAL COMMENTS	The authors have been responsive to the reviewer comments and have substantially revised the paper in response. I have no further comment
---

VERSION 2 – AUTHOR RESPONSE

Reviewer 1. I was glad to hear that the revision resulted in a more comprehensive and clear paper. Thanks for your suggestions and help.

Reviewer 2. Thanks for your detailed help in improving this manuscript, and for accepting the revision.